# Landscape Heterogeneity Drives Genetic Diversity in the Highly Dispersive Moss *Funaria hygrometrica* Hedw.

**DOI:** 10.3390/plants13192785

**Published:** 2024-10-04

**Authors:** Mahmoud Magdy, Olaf Werner, Jairo Patiño, Rosa María Ros

**Affiliations:** 1Department of Plant Biology, Faculty of Biology, Murcia University, 30100 Murcia, Spain; werner@um.es; 2Genetics Department, Faculty of Agriculture, Ain Shams University, Cairo 11241, Egypt; 3Island Ecology and Evolution Research Group, Institute of Natural Products and Agrobiology, (IPNA-CSIC), 38206 Tenerife, Spain; jpatino@ipna.csic.es

**Keywords:** altitudinal gradient, cp *rps*3-*rpl*16 spacer, cosmopolitan bryophytes, rire moss population, nrITS sequence, haplotype diversity, mt *rpl*5-*rpl*16 spacer, genetic structure

## Abstract

*Funaria hygrometrica*, a cosmopolitan moss species known for its remarkable dispersal capacity, was selected as the focal organism to investigate the relationship between landscape features and genetic diversity. Our study encompassed samples collected from two distinct regions: the Spanish Sierra Nevada Mountains (SN), characterized by a diverse landscape with an altitudinal difference of nearly 3500 m within a short distance, and the Murcia Region (MU) in Southeast Spain, characterized by a uniform landscape akin to the lowlands of Sierra Nevada. Genotyping analysis targeted three genetic regions: the nuclear ribosomal internal transcribed spacer (nrITS), the chloroplast *rps*3-*rpl*16 region, and the mitochondrial *rpl*5-*rpl*16 spacer. Through this analysis, we aimed to assess genetic variability and population structure across these environmentally contrasting regions. The Sierra Nevada populations exhibited significantly higher haplotype diversity (Hd = 0.78 in the highlands and 0.67 overall) and nucleotide diversity (π% = 0.51 for ITS1) compared to the Murcia populations (Hd = 0.35, π% = 0.14). Further investigation unveiled that samples from the lowlands of Sierra Nevada showed a closer genetic affinity to Murcia than to the highlands of Sierra Nevada. Furthermore, the genetic differentiation between highland and lowland populations was significant (*Φ*_ST_ = 0.55), with partial Mantel tests and ResistanceGA analysis revealing a strong correlation between ITS1-based genetic diversity and landscape features, including altitude and bioclimatic variables. Our study elucidated potential explanations for the observed genetic structuring within *F. hygrometrica* samples’ populations. These included factors such as a high selfing rate within restricted habitats, a limited average dispersal distance of spores, hybrid depression affecting partially incompatible genetic lineages, and recent migration facilitated via human activities into formerly unoccupied areas of the dry zones of Southeast Spain.

## 1. Introduction

Bryophytes, including mosses, exhibit broader distribution ranges compared to vascular plants, with low levels of endemism [1,2,3,4]. This widespread distribution is attributed, in part, to the exceptional dispersal capacity of windborne spores, as documented by Muñoz et al. [5], Shaw et al. [6], and Gibby [7]. Notably, mosses, such as *Funaria hygrometrica* Hedw., often demonstrate high dispersal abilities [8,9]. For instance, minimal differentiation was recorded among three bryophyte species with an intercontinental Mediterranean disjunct distribution pattern [10]. Similarly, genetic diversity in the cosmopolitan moss *Ceratodon purpureus* (Hedw.) Brid. was linked to long-distance migration within the Northern Hemisphere and Australasian regions, with more frequent migration observed between these regions relative to equatorial populations [11]. *Bryum argenteum* Hedw., another cosmopolitan moss, showed limited intercontinental genetic differentiation, except for the Antarctic continent [12].

Previous studies have indicated that cosmopolitan bryophytes often exhibit a wide range of habitat preferences, primarily due to their phenotypic plasticity, rather than the development of distinct ecotypes [13,14]. However, recent investigations have unveiled a genetic structure within *B. argenteum* populations inhabiting the Spanish Sierra Nevada Mountains, with specific haplotype clades restricted to elevations above 2000 m [15]. This genetic differentiation is likely influenced by the region’s unique climatic conditions, in which sporophyte formation is rare, reducing dispersal through gametophyte fragments. In other high-elevation ecosystems, such as that of Tibet, moss species like *Didymodon rigidulus* have shown significant variations in morphological traits, such as plant height and leaf size, in response to harsh environmental conditions [16]. These adaptations are primarily driven by temperature-related factors, including the mean annual temperature and potential evapotranspiration, rather than low precipitation or drought.

*Funaria hygrometrica* (Family Funariaceae, Order Funariales) is characterized as an acrocarpous, autoicous, and protandrous moss with an annual life cycle [4,17]. It thrives in shaded, damp soil and is commonly found on walls, the crevices of rocks, and recently burnt areas [3]. While its distribution pattern is primarily circumpolar wide–temperate, *F. hygrometrica* is considered cosmopolitan due to its presence in northern and temperate latitudes throughout the Holarctic, albeit infrequently in tropical and Southern Hemisphere temperate zones [18]. The species produces abundant, long-lasting, small spores that can be dispersed over long distances due to various factors, including wind, rain, insects, and human activities, with a peculiar morphological aspect (Figure 1).

The genetic variability of *F. hygrometrica* has been extensively studied, particularly in relation to its adaptability to environmental stresses and its evolutionary history. Previous research has identified chromosomal diversity and polyploidy events that have shaped the species’ evolutionary trajectory [19]. Studies examining the response of repetitive non-coding DNA to heavy metal stress have highlighted the potential for stress-induced genomic modifications [20]. Genomic scanning approaches have also been used to detect loci under selection along climate gradients, linking genetic variation to environmental factors [21]. Further investigations into the species’ capacity for bioremediation have demonstrated its effectiveness as a lead (Pb) adsorbent, emphasizing its genetic adaptability in response to anthropogenic pollutants [22]. Transcriptomic analyses of *F. hygrometrica* and related species have revealed divergent developmental pathways [23], while studies exploring hormonal regulation have provided insights into the role of auxin in growth and development [24]. Collectively, these studies highlight the genetic plasticity of *F. hygrometrica* and its ability to adapt to a range of environmental challenges.

In a broadly distributed species like *F. hygrometrica* with high dispersal capabilities, several questions arise: (1) Does genetic diversity vary between complex and uniform landscapes of comparable extension? (2) Is genetic structure correlated with ecological conditions across different altitudes in a diverse landscape? (3) Does genetic variation in a uniform part of a diverse landscape resemble that of a uniform landscape at a significant geographic distance? To address these questions, we employed a multi-locus sequencing approach, encompassing nuclear (nrITS), chloroplast (*rps*3-*rpl*16), and mitochondrial (*rpl*5-*rpl*16) markers due to their proven efficacy in phylogeographic and population genetic studies [12,15,25,26,27,28] to assess variation within *F. hygrometrica* populations sampled from the Spanish Sierra Nevada Mountains (SN) and nearby coastal regions. These results were contrasted with samples from the more uniform lowlands of the Murcia Region (MU), situated approximately 300 km northeast of the Sierra Nevada Mountains, a place where alpine summits that are snow-covered until summer are within close proximity.

## 2. Results

### 2.1. Genetic Variability, DNA Polymorphism, Haplotype Frequency, and Geographical Distribution

The hierarchical sampling strategy defined locations into three major regions, two lowlands, the MU and SN-LL regions, and one highland SN-HL region. Within the SN regions, the locations were grouped into four altitudinal zones (Z1–Z4), while different location groups from Murcia were grouped into three sampling areas (A1–A3; more details are given in Section 4.1). The four genotyped loci, namely nrITS1, nrITS2, cp *rps*3-*rpl*16, and mt *rpl*5-*rpl*16, were successfully amplified in the majority of samples for both SN and MU locations. Comprehensive data on DNA polymorphism, genetic parameters, and statistical analysis for each genotyped locus and region are presented in Table 1. Consistently, all markers exhibited haplotypes characterized by two distinct length (bp) groups (Groups A and B), displaying high pairwise identity within each group and lower pairwise identity between them (excluding gaps for these calculations). Indels were determined to be neutral based on Tajima’s D (*p* > 0.05) for all markers. However, considering the complete sequence of the four loci (separating the nrITS into nrITS1 and nrITS2), only nrITS2 exhibited neutrality and demographic equilibrium, prompting separate analyses of nrITS1 and nrITS2 to mitigate the potential effects of incomplete lineage sorting (ILS), different evolutionary histories, or concerted evolution.

#### 2.1.1. Nuclear ITS1

For the ITS1 locus, an analysis across the SN highland (SN-HL), over 1300 m above sea level (a.s.l.), SN lowland (SN-LL, below 1300 m a.s.l.), and MU populations revealed varying levels of genetic diversity and differentiation. In the SN-HL region, a total of 22 sequences were identified, exhibiting a high haplotype diversity (Hd) of 0.78 and a moderate nucleotide diversity (π%) of 0.51. Group A sequences, characterized by lengths of 244–249 base pairs (bp), demonstrated a pairwise identity of 98.70%. In contrast, the SN-LL region presented Group A sequences of 254–255 bp with a higher pairwise identity of 99.70% among 14 sequences. Meanwhile, in the MU region, 40 sequences were recorded, indicating a lower Hd of 0.35 and π% of 0.14, with Group A sequences at 248 bp showing a pairwise identity of 99.70% among 40 sequences. When considering the total alignment, comprising 124 sequences, a moderate Hd of 0.67 and π% of 0.40 were observed, suggesting substantial genetic diversity. The Tajima’s D value of −1.83 * indicates a departure from neutral evolution, possibly due to demographic processes or selection pressures (Table 1).

The most abundant haplotype of the 25 found, nr1-03, was found in 73 samples (37 in SN and 36 in MU), and it exhibited a widespread geographical distribution across all areas and zones (details about the areas’ and zones’ definitions are provided in Section 4.1). Haplotype nr1-09, present in 12 samples, was exclusive to the four zones of SN. A minimum spanning network defined two haplotype groups based on sequence length: Group nr1-A (244–249 bp), encompassing haplotype nr1-03 and others, and Group nr1-B (254–255 bp), comprising haplotypes nr1-09, 19, and 22, with haplotype nr1-02 positioned apart from both groups (Figure 2a).

#### 2.1.2. Nuclear ITS2

For the ITS2 locus, the examination of genetic variation across the SN-HL, SN-LL, and MU populations unveiled distinct patterns. In the SN-HL region, 22 sequences were identified, exhibiting a moderate Hd of 0.37 and a relatively high π% of 0.88. The Group A sequences, at 311–312 bp, displayed a pairwise identity of 93.00% among 17 sequences. Conversely, in the SN-LL region, the Group A sequences of 316 bp demonstrated a higher pairwise identity of 99.99% among 67 sequences. Meanwhile, the MU region revealed Group B sequences at 316 bp with a pairwise identity of 99.70% among 37 sequences. Overall, the total alignment of 122 sequences displayed a moderate Hd of 0.31 and π% of 0.75, indicative of substantial genetic diversity. The Tajima’s D value was insignificant at −1.74 (Table 1).

The predominant haplotype of the 10 haplotypes that were found, nr2-01, was detected in 101 samples (66 in SN and 35 in MU), and it exhibited a broad geographical distribution across all areas and zones. Two haplotype groups were identified based on sequence length in the minimum spanning network: Group nr2-A (316 bp), including haplotype nr2-01 and others, and Group nr2-B (311 and 312 bp), encompassing haplotypes nr2-03, 04, and 05 (Figure 2b).

#### 2.1.3. Chloroplast *rps*3-*rpl*16

An analysis of the cpDNA locus across the SN-HL, SN-LL, and MU populations provided insights into genetic diversity and differentiation. In the SN-HL region, 22 sequences were identified, displaying a high Hd of 0.72 and a moderate π% of 0.20. The Group A sequences at 778–779 bp exhibited a pairwise identity of 99.80% among 12 sequences. Conversely, in the SN-LL region, the Group A sequences of 794–798 bp demonstrated a higher pairwise identity of 99.50% among 67 sequences. Meanwhile, the MU region revealed Group A sequences at 795–797 bp with a pairwise identity of 99.90% among 20 sequences. The total alignment of 100 sequences displayed a moderate Hd of 0.66 and π% of 0.16, suggesting considerable genetic diversity. The Tajima’s D value was significant at −2.18 (Table 1).

The most abundant haplotype of the 29 haplotypes that were found, cp-01, was found in 58 samples (41 in SN and 17 in MU), and it displayed a widespread geographical distribution across all zones. Two haplotype groups were discerned based on sequence length: Group cp-A (794–798 bp), containing haplotype cp-01 and others, and Group cp-B (778–779 bp), comprising haplotype cp-20 and others (Figure 2c).

#### 2.1.4. Mitochondrial *rpl*5-*rpl*16

An exploration of the mtDNA locus across the SN-HL, SN-LL, and MU populations revealed notable genetic diversity and differentiation. In the SN-HL region, 22 sequences were identified, displaying a moderate Hd of 0.43 and a low π% of 0.03. Group A sequences at 790–791 bp exhibited a pairwise identity of 99.98% among 66 sequences. In the SN-LL region, Group A sequences of 836 bp demonstrated a high pairwise identity of 99.90% among 18 sequences. Meanwhile, the MU region revealed Group A sequences at 792 bp with a pairwise identity of 100% among 51 sequences. The total alignment of 136 sequences displayed a moderate Hd of 0.65 and π% of 0.02, indicative of substantial genetic diversity. The Tajima’s D value was significant at −2.32 (Table 1).

The prevalent haplotype of the 13 found, mt-01, was identified in 62 samples exclusive to SN and distributed across all four zones. Two haplotype groups were delineated based on sequence length: Group mt-A (794–802 bp), containing haplotype mt-01 and others, and Group mt-B (778–779 bp), including haplotype mt-02 and others (Figure 2d).

### 2.2. Genotypic Diversity

All analyzed parameters indicated greater diversity within SN populations compared to MU populations. The pairwise identity of sequences was higher in MU than in SN. SN exhibited between 1.75 (nrITS2) and 5.5 (nrITS1 and mt*rpl*5-*rpl*16) times more haplotypes than MU. Furthermore, haplotype diversity and nucleotide diversity consistently showed higher values in SN compared to MU. To ascertain whether the observed higher diversity in SN was statistically significant, 10,000 bootstrap replicates of aligned sequences were generated, and the 95% confidence intervals of nucleotide diversity values were calculated. The results revealed significantly lower values observed for MU across three of the four markers. Apart from nrITS2, the 95% confidence intervals for MU and SN did not overlap (Appendix A).

Haplotypes from the four genotyped loci were typically segregated into two groups defined by length (A, with shorter sequences being more frequent, and B, with longer sequences being less frequent). These groupings were not solely determined by indels; even when gaps were excluded, the pairwise identity within each group remained markedly higher than among the groups. Combining the haplotype length groups of all four markers revealed 10 of the 16 theoretically possible combinations. The most prevalent combination (AAAA) occurred 56 times in SN and 18 times in MU, while the combination BBBB was observed only seven times in SN with no instances in MU (Table 2).

Samples from SN-LL (SN-Z1 and SN-Z2) exhibited a predominantly homogeneous state (i.e., AAAA or BBBB). Conversely, SN-HL samples (SN-Z3 and SN-Z4), showed increasing instances of non-homogeneity. Samples from SN-Z1, samples from SN-Z2, and some samples from other zones and areas displayed non-homogeneity either at the maternal lineage level (AAAB, AABA, and BBAB) or at the nuclear level (BAAA, ABAA, and BABB), with some samples exhibiting non-homogeneity at both levels (BABA and AABB). In contrast, all genotyped samples from MU were homogeneous (AAAA) within Group A, with a single incomplete sample displaying a Group B mtDNA haplotype (sample MU_12; Table 2).

### 2.3. Geographical vs. Altitudinal Distribution of Haplotype Groups

Based on haplotype groups A and B, the genetic combination distribution was examined with respect to geographical and altitudinal conditions for each marker. Samples were categorized into SN-LL, SN-HL, and MU populations. Pairwise comparisons revealed a non-random distribution of haplotype groups, particularly significant differences that were observed when comparing SN-HL with SN-LL or MU. Conversely, differences between SN-LL and MU were not significant (Table 3).

### 2.4. Genetic Differentiation, Migration, and Linkage Disequilibrium

#### 2.4.1. Genetic Differentiation and Population Diversity

We computed AMOVA, *Φ*_CT_, *Φ*_SC_, and *Φ*_ST_ to assess genetic differentiation. When considering the geographical factor (SN vs. MU), *Φ*_CT_ (between regions) showed non-significance for all genotyped loci, indicating that differences between SN and MU are not substantial, unlike *Φ*_SC_ (within zones/areas) and *Φ*_ST_ (among sampling locations), which exhibited significant values (0.05 < *p* < 0.001), except for *Φ*_SC_ of nrITS2 (*p* > 0.05). When populations were stratified by altitude (MU and SN-LL vs. SN-HL), both *Φ*_CT_ and *Φ*_ST_ were higher and significant for all genotyped loci (*p* ≤ 0.001), unlike *Φ*_SC_, whose values were insignificant except for *Φ*_SC_ of cpDNA (0.05 < *p* < 0.001). Significant *Φ*_ST_ values were observed in both approaches, ranging from 0.15 to 0.55, with the mtDNA spacer being the sole marker significant at all hierarchical levels (Table 4).

#### 2.4.2. Migration and Genetic Diversity

Migration rates (M) were estimated through a Bayesian inference of historical migration analysis, assumed to be constant across all four genotyped loci (Figure 3). First, the migration rates were calculated between SN (including both highland and lowland zones) and MU. The migration from MU to SN was notably high (M = 1919) compared to the migration from SN to MU (M = 77). Second, the migration rates were calculated between the three major regions, SN-HL, SN-LL, and MU. M values among the three major regions revealed higher rates from MU to SN-LL than to SN-HL (M = 1705 and 1115, respectively). SN-HL exhibited a higher migration rate to SN-LL (M = 317) than to MU (M = 133), with SN-LL recording the lowest migration rates overall. Additionally, M values to SN-HL were higher than those to MU (M = 292 and 41, respectively). Theta values (Θ), indicating genetic diversity, were higher in SN than in MU (Θ = 0.06 and 0.01, respectively), and they were particularly slightly higher in the lowlands than in the highlands (Θ = 0.05 and 0.04, respectively). SN-LL showed higher Θ values due to cpDNA variation, while nrITS1 and mtDNA exhibited higher values in SN than in MU. NrITS2 displayed the highest Θ values for both SN and MU.

#### 2.4.3. Linkage Disequilibrium Analysis

The linkage disequilibrium (LD) test for the four sequenced loci revealed significant disequilibrium values for all combinations of loci except for nrITS1–cpDNA. This indicates that loci are not independent but are associated in a non-random manner. Specifically, certain combinations of haplotypes of different loci were more frequent than expected in a random process. Only the nrITS1 spacer with the cpDNA region displayed values slightly above the significance level of 0.05, with a value of 0.06. When LD in SN samples below 1300 m a.s.l. were studied, all combinations of loci exhibited disequilibrium (*p* = 0.00). However, when samples above 1300 m a.s.l. were considered, only nrITS1 and nrITS2 showed a non-random association, which was not unexpected, considering that the two loci are separated by the short 5.8S rRNA gene and arranged in tandem repeat units (Table 5).

### 2.5. Landscape Heterogeneity in Sierra Nevada Mountains

The partial Mantel test was applied to detect the association between the pairwise genetic distance matrices of sequenced regions and the Euclidean distance matrices of climatic and bioclimatic variables while controlling for geographical distance (details in Section 4.5). During the presumed life cycle of *F. hygrometrica* from SN, the estimated average maximum temperature (Tmax) Euclidean matrix demonstrated a significant correlation (*r*-value = 0.52, *p* ≤ 0.01) with the nrITS1 genetic distance matrix. Similarly, a Euclidean matrix of the interaction between drought periods and warm temperatures throughout the year (the precipitation in the driest month, the precipitation in the driest quarter, and the precipitation in the warmest quarter) exhibited a significant correlation (*r*-value = 0.38, *p* ≤ 0.01) with the mtDNA genetic distance matrix. Conversely, the cpDNA and nrITS2 genetic distance matrix displayed insignificant correlations with all tested environmental factors (*p* > 0.05). Moreover, all markers showed insignificant correlations with the geographical distance matrix (*p* > 0.05).

PCA analysis clustered bioclimatic variables into two groups and two singletons, each tested for their association with the genetic structure of sampled *F. hygrometrica* populations (Appendix A). In ResistanceGA analysis, altitude, geographical distance, land cover, and human population density are crucial variables considered to assess how landscape features influence genetic connectivity among populations [29]. These factors are integrated into resistance surfaces to model barriers or facilitators of gene flow, revealing how natural and anthropogenic landscapes shape genetic structure and inform conservation strategies. When the ResistanceGA analysis of all loci combined (regardless of neutrality status) was applied, PCA Group 2 and the two singletons were deemed insignificant, whereas genetic diversity was associated with the average annual temperature range from PCA Group 1. When each locus was tested separately, only ITS1 exhibited a statistically significant association with a bioclimatic variable (precipitation in the wettest month), with less likelihood of an association with the annual temperature range (Figure 4). However, the effects of altitude, geographical distance, land cover, and human population density on the variation in *F. hygrometrica* populations in SN were not evident.

## 3. Discussion

The main findings of the current work demonstrate how an organism with high dispersal capacity inhabiting a diverse landscape with an altitudinal gradient can harbor significantly higher genetic diversity than in a uniform landscape of the same extension. Despite the relatively small size of the sampling area in the Sierra Nevada Mountains, an unexpectedly high genetic variability was found in *F. hygrometrica*. Individuals sampled from this mountain range showed significantly higher haplotype and nucleotide diversity than those of the Murcia Region, which supports our first hypothesis that genetic diversity is higher in a diverse landscape with an altitudinal gradient. The cosmopolitan moss *B. argenteum* is another species that was studied in the Sierra Nevada Mountains, and it shows very high genetic diversity. In this species, diversity was clearly structured according to altitude [15]. The extremely diverse landscape with the altitudinal gradient of the Sierra Nevada Mountains accommodates high genetic diversity; therefore, this mountain range possesses not only rare and endemic species [30,31] but also an important part of a hidden diversity of frequent and widely distributed species [12,15,21,25,26].

High genetic differentiation among the diverse landscapes was detected, even though statistical evidence for the altitudinal gradient to explain the estimated genetic variation was not detected using the altitude or geographic distance resistance layers. The higher genetic differentiation that was found via AMOVA when the grouping approach was changed from geographical to altitudinal (especially in the case of the *Φ*_CT_ values that compare regions) reflected the effect of altitude or correlated factors, rather than geographical distribution, on the genetic differentiation among the sampled populations (Table 4). That would support the second hypothesis of the introduction, namely that genetic structure is related to altitude in *F. hygrometrica.* This is in line with the work of Korpelainen et al. [32], who detected a significant relation between vertical and genetic distance along altitudinal gradients in the moss *Pleurozium schreberi* (Willd. ex Brid.) Mitt. But in their case, no sporophytes were observed in the sampled populations, and dispersal was, therefore, probably limited due to vegetative propagation with a reduced dissemination range. Shaw [33,34] reported genetic differences in *F. hygrometrica* populations growing on uncontaminated and heavy metal-contaminated soils based on growth rates. These data suggest that there exist genetic varieties capable of growing in extreme environments that are distinct from plants growing on non-contaminated soil. Additionally, transcriptomic studies comparing *F. hygrometrica* and *Physcomitrella patens* have highlighted species-specific genetic adaptations that contribute to divergent development and environmental resilience [23]. These studies, together with our findings, underscore the genetic plasticity of *F. hygrometrica*, particularly in diverse landscapes like the Sierra Nevada, where altitude plays a significant role in structuring genetic diversity.

The results also seem to confirm our expectations that the samples of the Sierra Nevada Mountains lowlands would be more closely related to the Murcia Region than to the Sierra Nevada Mountains highlands (the third hypothesis). This can be seen from the distribution of the length classes of all markers, the results of the AMOVA, and the LD data (Table 4 and Table 5). All markers showed two length groups. Group A, with longer sequences, always included the more abundant haplotypes, and Group B, with shorter sequences, always included the less abundant haplotypes. Both groups were separated by a pronounced genetic distance. In the Sierra Nevada Mountains, comparative observations between all sequenced loci showed that some samples contained marker haplotypes of different groups. These samples are referred to as non-homogeneous genotypes (Table 2). Somewhat surprisingly, some samples were found to be non-homogeneous at the maternal DNA level because, generally, it is assumed that both chloroplasts and mitochondria are inherited exclusively through the maternal line in bryophytes [35]. Some researchers question the generalized case of the maternal inheritance of mitochondria and chloroplasts in plants [36,37,38,39].

Another fact that suggests that altitude or a correlated factor is important to explain the genetic structure observed in *F. hygrometrica* is that the haplotype length groups were non-randomly distributed when comparisons between the Sierra Nevada Mountains lowlands vs. Sierra Nevada Mountains highlands and the Sierra Nevada Mountains highlands vs. the Murcia Region were made (Table 4). The non-homogeneous genotypes are mostly present above 1300 m a.s.l., in what we called the hybrid zone among both types. In the hybrid zone, homogeneous genotypes of the most abundant haplotype combination (AAAA) were found to decrease above 1300 m a.s.l., in contrast to homogeneous genotypes of the less abundant haplotype combination (BBBB) and the non-homogeneous genotypes (Table 2). Although the inspection of the length-group data suggested a recombination between different lineages (Table 1 and Figure 2), analyses of LD clearly showed that there is a non-random association between genotypes of the different sequenced loci (Table 5). Only the Sierra Nevada Mountains highlands do not show LD, except for the nrITS1 and nrITS2 loci, which are genetically linked through proximity (Table 5). A preferential association of independent loci may be the result of various factors [40]. According to these authors, mutations are only important in regions without recombination, as is the case for sex chromosomes; otherwise, recombination leads to a rapid decay of disequilibrium. Random drift is only important in populations with a small relative population size and loci with a low recombination rate; this is the case for sex chromosomes or regions with chromosomal inversions [40]. In our case, nuclear DNA, on the one hand, and mitochondrial and chloroplast DNA, on the other hand, should recombine freely. In the studied regions, the population size of *F. hygrometrica* is not low, so random drift is probably not involved. Inbreeding can lead to considerable disequilibrium, essentially because it slows the decay process down. In bryophytes, antheridia and archegonia develop on the haploid gametophyte. Therefore, self-fertilization in monoic bryophytes leads to completely homozygous sporophytes, and all spores are genetically identical [41]. Outcrossing is restricted due to the limited distance of sperm movement in water, which is in the range of cm [42] or, in some cases, up to more than one meter [43]. Therefore, most of the reproduction occurring in *F. hygrometrica* might indeed be clonal despite being the result of an apparently sexual mechanism. The degree of self-fertilization in *F. hygrometrica* is currently unknown [44]. The fact that disequilibrium occurs at lower altitudes (below 1300 m a.s.l.) but not in the locations situated at higher altitudes (above 1300 m a.s.l.; Table 5) indicates that outcrossing must occur to some degree, at least at higher altitudes, perhaps favored by the higher availability of water since the precipitation of the wettest month variable was identified as a significant predictor of genetic differentiation among the regions.

The detected level of migration might be due to human-related factors (e.g., Wen and Hsiao [45]) or historical migrations associated with past climate change (e.g., the glacial period [46]). The presence of human beings in the studied region in evolutionarily recent times could have increased the number of suitable habitats for this species (e.g., along roadsides), and this development would have caused a population expansion. Population expansion could at least partially explain the significant negative values for Tajima’s D. Consequently, the high genetic diversity found in the Sierra Nevada Mountains might simply be the result of a high migration rate into a landscape formed due to human factors. If this is the case, a uniform landscape of a similar size and with a similar or an even higher human impact—but with uniform environmental conditions—should have a high genetic diversity as well, which was not the case in the studied uniform landscape (Murcia Region). The analysis using Migrate-N (Figure 3) suggests that there is indeed migration from the Murcia Region in the direction of the Sierra Nevada Mountains, but this cannot explain the high diversity found in the Sierra Nevada Mountains, as the haplotypes found especially in the highlands of the Sierra Nevada Mountains are almost completely absent from the Murcia Region. Alternatively, one might speculate that, in the Sierra Nevada Mountains, some populations in more conserved habitats are present, which are limited to more natural habitats. But all samples were taken from very similar places (road embankments), keeping the sampling conditions for all samples as similar as possible. However, the high genetic diversity in the Sierra Nevada Mountains might reflect the existence of formerly isolated populations that survived under the more humid conditions of this high mountain range, while the harsh and dry conditions in southern–eastern (SE) Spain did not allow the establishment of populations of *F. hygrometrica* in most other parts of this region. Only human-made changes (especially the construction of roads) allowed the migration of this species into unoccupied territories in SE Spain. Under this scenario, migrants finally came in contact with the natural and genetically distinct populations in the Sierra Nevada. When the past climate-change effect on the migration of this species is considered, the recolonization of wide areas in the Sierra Nevada Mountains after glaciation is not an important factor in the population history or the high genetic differentiation of *F. hygrometrica* since only altitudes above 2000 m a.s.l. were clearly covered by glaciers in the Sierra Nevada Mountains [47]. Moreover, the average annual temperature was 12.7 °C at sea level [48], which is higher than the values reported for any German weather station at present [49].

An alternative explanation is that the reproductively isolated populations of *F. hygrometrica* developed elsewhere and came in contact afterwards in the Sierra Nevada Mountains region. In the case of secondary contact, it is possible that pre- or postzygotic isolation mechanisms prevent gene flow and recombination to some degree. In the moss species *C. purpureus,* McDaniel et al. [50,51] observed that interpopulation hybrids showed segregation distortion and developmental abnormalities. In *F. hygrometrica*, previously published AFLP (amplified fragment-length polymorphism) data from the Sierra Nevada Mountains region suggest the presence of two lineages that could be interpreted as cryptic species [21]. However, the sequence data presented here combining nuclear, mitochondrial, and chloroplast markers show that there was no clear reproductive isolation between the lines examined from the sampled areas/zones (Figure 2). Therefore, possible restrictions to recombination between different lineages are not absolute. But in plants, hybridization between clearly distinct species is a frequent phenomenon [52]. As a consequence, we cannot totally exclude the possibility that the samples of *F. hygrometrica* collected from the Sierra Nevada Mountains belong to two cryptic species, which came into secondary contact.

Taken together, there are four not mutually exclusive explanations for the apparent contradiction between the high dispersal capacity of spores and genetic structuring according to ecological conditions within a short range: (1) a high selfing rate for locally adapted lineages, (2) a low average dispersal distance of spores limited by environmental conditions, (3) hybrid depression affecting partially incompatible genetic lineages, and (4) the recent migration of *F. hygrometrica* into formerly unoccupied areas of the dry zones of SE Spain, favored by human activities and the contact of migrants with the “island” populations in the Sierra Nevada Mountains, resulting in high genetic diversity in the Sierra Nevada Mountains region.

## 4. Materials and Methods

### 4.1. Sampling Regions, Plant Material, and Environmental Conditions

The Sierra Nevada range is situated in the Granada and Almeria provinces of Andalusia, southern Spain (Figure 5a–c). It boasts over 20 peaks exceeding 3000 m a.s.l., with the Mulhacen peak reaching the highest point on the Iberian Peninsula at 3479 m a.s.l. Designated as a biosphere reserve in 1986, portions of the range were later declared a National Park in 1999, while others were designated as a Natural Park. Climatic data from the Worldwide Bioclimatic Classification System (1996–2017) [53] indicate that Motril, near the Mediterranean coast, experiences an average annual temperature of 18.1 °C with annual precipitation of 455 mm, while Pradollano, at 2507 m a.s.l., records an average annual temperature of 3.9 °C and 693 mm in annual precipitation. Notably, summers in the SN are exceedingly dry, with little to no precipitation in July and August (Figure 6a). The region has been subject to numerous studies due to its rich biodiversity resulting from unique ecological conditions over millennia [15,21,25,26,30,31,54,55,56]. MU lies in southeastern Spain, bordered by the Mediterranean Sea to the south (Figure 5a,b,d). The portion of MU included in this study experiences average annual temperatures of around 17 °C (17.3 °C in Cartagena, situated on the coast at 14 m a.s.l., and 16.5 °C in Totana at 200 m a.s.l. inland), with low average annual precipitation close to 300 mm across all studied areas [53]. Similar to SN, precipitation in MU is scarce during the summer months (Figure 6b).

*Funaria hygrometrica* samples were collected from diverse environments across SN, ranging from the Mediterranean coast to alpine summits (Perimeter~190 km and sampling area~700 m^2^), whereas sampling in MU was confined to the dry lowlands below 350 m a.s.l. (perimeter~38 km and sampling area~1200 km^2^). To maintain ecological consistency across sampling areas (with exceptions for climatic conditions), plants were sourced from road embankments and similar sites in both regions. In total, 84 samples from 17 locations were collected from SN (Appendix A). These locations were categorized into four altitude-based zones. Zone 1 (SN-Z1) encompassed seven locations distributed between 10 and 647 m a.s.l., Zone 2 (SN-Z2) comprised four locations situated between 755 and 1328 m a.s.l., Zone 3 (SN-Z3) included two locations ranging from 1644 to 1666 m a.s.l., and Zone 4 (SN-Z4) consisted of four locations spanning altitudes from 2156 to 2683 m a.s.l. For some analyses, SN-Z1 and SN-Z2 were combined as the SN lowlands, contrasting with SN-Z3 and SN-Z4, considered the SN highlands.

Additionally, 53 samples were collected from 20 locations in MU (Appendix A), grouped by geographical proximity. To reflect the change in grouping criterion compared to SN (geography vs. altitude), the term “area” was used instead of “zone” for MU. Area 1 (MU-A1) comprised six locations situated inland near the town of Lorca (altitudes between 157 and 343 m a.s.l.), Area 2 (MU-A2) included ten coastal locations (altitudes between 2 and 139 m a.s.l.), and Area 3 (MU-A3) contained six locations near the city of Murcia (altitudes between 77 and 307 m a.s.l.).

### 4.2. In Vitro Cultivation, DNA Extraction, and PCR Amplification

Prior to cultivation, all plant samples were meticulously identified and confirmed using Nikon^®^ stereomicroscopes and Olympus microscopes through a process adhering to morphological descriptions available in the literature [58], and they were deposited into the Murcia University Herbarium collection (Appendix A). Moss spores were isolated from mature capsules (one capsule per sample) and cultured in vitro to obtain axenic material as described by Sabovljevic et al. [59] and Magdy et al. [21]. A mixed protonema from each sample was utilized for DNA extraction using the GenElute^TM^ Plant Genomic DNA Miniprep Kit (Cat# G2N350, Sigma-Aldrich, St. Louis, MO, USA) following the manufacturer’s protocol. Alternatively, DNA extraction was performed from the upper part of sterilized gametophores of samples lacking capsules or viable spores.

Several regions were targeted and sequenced to ensure adequate polymorphism, reflecting variation within the sampled locations. A nuclear DNA internal transcribed spacer (ITS) was amplified using modified primers from Douzery et al. [60] to match the published sequence of *F. hygrometrica* (GenBank: X74114). The cp *rps*3-*rpl*16 region, comprising the rpl16-rps3 spacer, rpl16 exon I, and rpl16 intron, was amplified using a new set of primers designed and optimized for amplification. The mt *rpl*5-*rpl*16 intergenic spacer was amplified using primer pairs according to Liu et al. [61] (Appendix A). Amplifications were performed using Green GoTaq^®^ Flexi DNA polymerase (Promega, M8295, Madison, WI, USA) in 50-µL reactions containing 50 ng of DNA, 1x GoTaq^®^ Flexi buffer, 4 mM of MgCl_2_, 0.20 mM of dNTPs (each), 1 µM of each primer, and 1.25U of Go Taq™. PCR cycling conditions were adjusted as specified in Table (S3). PCR products were visualized using 1.5 % (1.5 g per 100 mL) agarose gel electrophoresis stained with 1 x RedSafe^®^ (5 µL per 100 mL). Purified fragments were sequenced using the automated Sanger method (Secugen, Spain, and Macrogen, The Netherlands).

### 4.3. Genetic Variation, Diversity, and DNA Polymorphism Analyses

Chromatograms were assembled and edited using Bioedit [62] and Geneious Prime [63]. DNASP V5 [64] and ARLEQUIN V3.5 [65] were employed to estimate genetic variability indices, including sequence minimum and maximum lengths, the pairwise genetic identity percentage (PI), the number of variable sites, the number of haplotypes (h), the number of indels (insertion–deletion sites), the number of indel-haplotypes, haplotype diversity (Hd), and Tajima’s D neutrality for haplotypes and recorded indels [66]. The mean nucleotide diversity (π% excluding gaps) and its standard deviation at 10,000 bootstraps were calculated using PEGAS [67]. To assess the significance of observed genetic diversity differences between regions, PEGAS was utilized to simulate 10,000 bootstrap replicates of the sequences and calculate π-values for each replicate. The 95% confidence interval (0.025–0.975) was determined using R 3.3.1 [68].

### 4.4. Genetic Differentiation and Migration

PopART (http://popart.otago.ac.nz, accessed on 1 October 2023) was employed to construct minimum spanning networks and assess population differentiation via an analysis of molecular variance, AMOVA. PhiCT (*Φ*_CT_) was calculated among regions (SN and MU); PhiSC (*Φ*_SC_) was determined within altitudinal zones/geographical areas (four altitudinal zones of SN and three geographical areas of MU), and PhiST (*Φ*_ST_) was calculated among sampling locations. Additionally, migration patterns between SN and MU, and among major geographical regions (SN-HL, SN-LL, and MU), were confirmed using Migrate-N [69], based on the four genotyped loci as a single dataset. Bayesian inference was employed for the search strategy with 1000 recorded steps in the chain and a full migration model. Estimated migration ratios were visualized using Circos 0.69 [70].

Pearson’s chi-square test [71] and Fisher’s exact test [72] were utilized to evaluate the frequency of haplotype-group occurrences at different altitudes (SN highlands vs. SN lowlands) or geographical regions (MU vs. SN) using the gmodels package of R [73].

Among SN’s altitudinal groups, an LD analysis was performed using ARLEQUIN V3.5 to test the non-random association between sequenced regions. The different alleles of the studied regions were coded at the haplotype level as standard data, with the gametic phase indicated as known. nrITS1 and nrITS2 were treated as different loci according to RDP4, v.4.56 [74], due to a recombination event between the two parts of the nrITS region. The significance level was set to *p* = 0.05.

### 4.5. Landscape vs. Genetic Diversity

The population genetic diversity of *F. hygrometrica* in SN was correlated with the spatial patterning of climatic and bioclimatic variation. Climatic and bioclimatic layers were obtained from the WorldClim database, version 1.4 [75], at 30 arc-second resolutions. Data from February to June were considered to encompass the approximate life cycle of *F. hygrometrica* in the studied area. Climatic and bioclimatic variables were standardized using z-score transformation and subjected to principal component analysis using PCO v3 software [76].

Partial Mantel tests were performed using GenAlEx V6 [77] to measure the association between the pairwise genetic distance matrices of sequenced regions and the Euclidean distance matrices of climatic and bioclimatic variables while controlling for geographical distance. The Mantel test was conducted with default pre-set values and parameters (9999 permutation steps).

To assess the association between landscape features (altitude, landcover, population density, climatic, and bioclimatic variables) and genetic differentiation, Ne’s standard genetic distance (Ds) among individuals using Populations 1.2.31 was estimated for each locus and for all loci combined. Landscape layers were converted into resistance raster layers and tested for association using ResistanceGA in R [29], allowing for the independent optimization of each landscape layer with a maximum iteration of 100.

## 5. Conclusions

In conclusion, this research sheds light on the genetic diversity, migration patterns, and landscape (specifically altitude and bioclimatic variables) influences on *Funaria hygrometrica* populations in the Sierra Nevada Mountains and the Murcia Region of Spain. The study revealed significant genetic differentiation among populations, with higher diversity observed in the diverse landscape with altitudinal gradient of the Sierra Nevada Mountains compared to the uniform landscape of the Murcia Region. Altitude appeared to play a crucial role in shaping the genetic structure, with populations at different altitudinal levels exhibiting distinct genetic profiles. Moreover, migration analysis highlighted a substantial gene flow between regions, with evidence suggesting both natural dispersal and human-mediated transport. Climate data underscored the environmental heterogeneity of the Sierra Nevada Mountains, likely contributing to the observed genetic diversity, especially the average maximum temperature and the interaction between drought periods and warm temperatures throughout the year. Overall, this research enhances our understanding of how landscape features (altitude and climate) influence the genetic structure and migration dynamics of bryophyte populations, providing valuable insights into the molecular adaptability to climatic changes in bryophytes, as well as the conservation and management strategies in these ecologically diverse regions.

## Figures and Tables

**Figure 1 plants-13-02785-f001:**
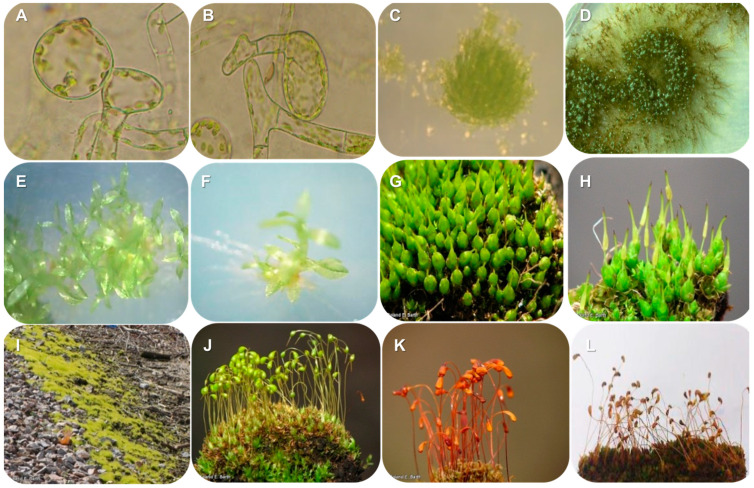
*Funaria hygrometrica* in different states of its life cycle. (**A**–**F**) In vitro cultivation and (**G**–**L**) fully matured plants. Images (**A**–**F**,**L**) were captured by the authors, while the remaining images were obtained from Roland E. Barth (Nature Centers of Bellevue, Nebraska) and used with permission.

**Figure 2 plants-13-02785-f002:**
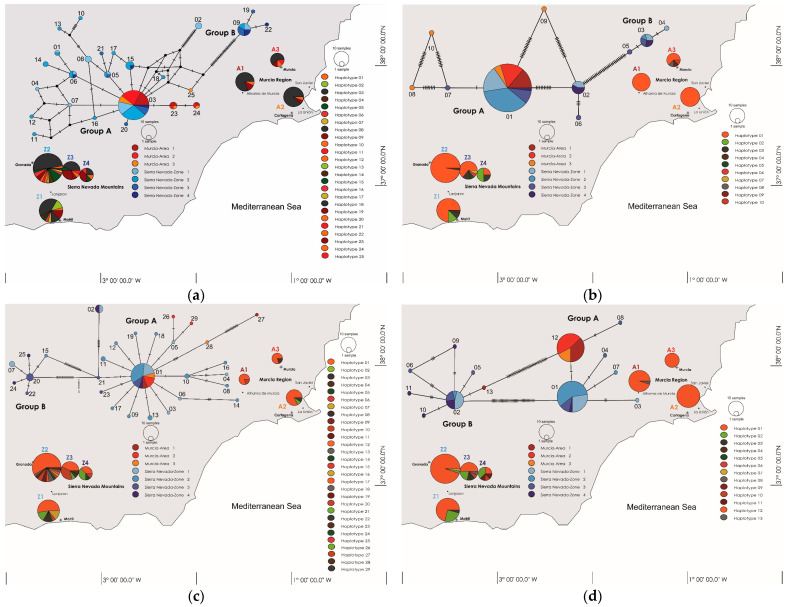
Minimum spanning network, geographical distribution, and haplotype frequencies of *F. hygrometrica* populations in southern Spain. The Murcia Region is delineated into three geographical areas (A1, A2, and A3), whereas the Sierra Nevada Mountains are subdivided into four zones (Z1, Z2, Z3, and Z4), arranged by altitude, ranging from 24 to 2700 m above sea level (a.s.l.). Samples from Z1 and Z2 represent the Sierra Nevada Mountains Lowlands (SN-LL), while samples from Z3 and Z4 represent the Sierra Nevada Mountains Lowlands (SN-LL). Haplotypes are classified into two groups based on their lengths: nr1-A and nr1-B for nrITS1 (**a**), nr2-A and nr2-B for nrITS2 (**b**), cp-A and cp-B for cp *rps*3-*rpl*16 (**c**), and mt-A and mt-B for mt *rpl*5-*rpl*16 (**d**).

**Figure 3 plants-13-02785-f003:**
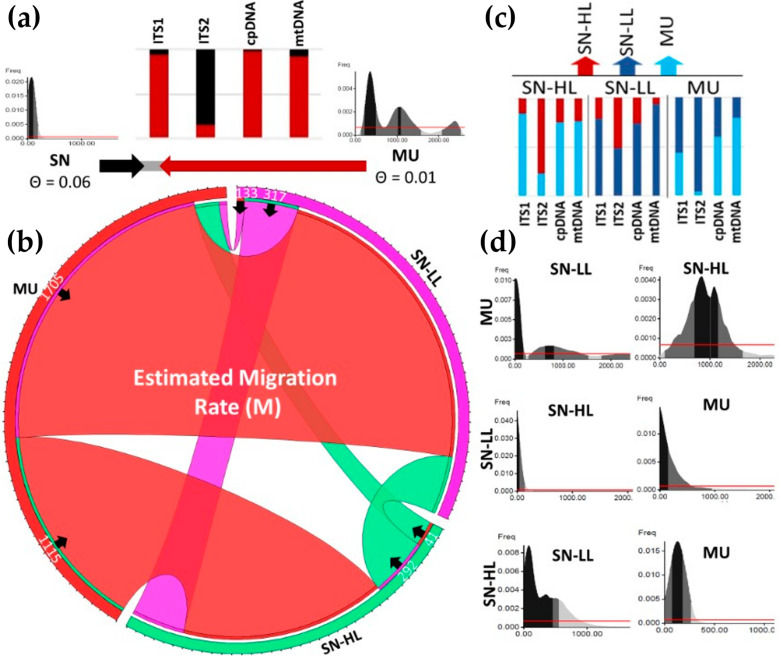
Migration rates across geographical and altitudinal groups of *F. hygrometrica* populations in the Sierra Nevada Mountains and Murcia Region, southern Spain. The graphical illustration depicts migration rates (M) estimated using the Migrate-N software v4.2.14, with sampling locations grouped into three geographical groups: Sierra Nevada Mountains (SN), divided into lowlands (up to 1300 m a.s.l., SN-LL) and highlands (above 1300 m a.s.l., SN-HL), and the Murcia Region (MU). Migration rates were estimated pairwise, and they are indicated using black arrowheads denoting the direction of migration between regions. In panel (**a**), migration rates between SN and MU populations are shown alongside theta values (Θ, an estimator of genetic diversity). MU (red arrow) exhibits higher migration rates toward SN (black arrow) across all loci except ITS2. Panel (**b**) illustrates the altitudinal difference within SN, with SN-LL populations showing lower Θ values than SN-HL populations, while migration rates between them vary slightly. The migration from MU to SN-LL populations surpasses that to SN-HL populations, whereas SN-HL exhibits marginally higher migration rates to MU populations compared to SN-LL populations. The red bars present MU populations, the green bars present SN-HL populations, and the violet bars present SN-LL populations. Panel (**c**) provides detailed migration rates by locus, highlighting distinct patterns observed for ITS2 compared to other markers. Finally, panel (**d**) presents the posterior distribution for each estimated migration rate (M).

**Figure 4 plants-13-02785-f004:**
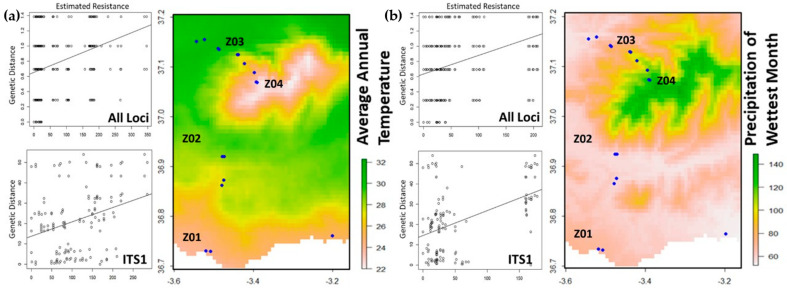
The analysis of genetic diversity in relation to landscape features. Bioclimatic variables, including average annual temperature range (**a**) and precipitation of the wettest month (**b**), exhibited the highest association coefficients and lowest AICc (corrected Akaike information criterion) values for all loci (combined) and ITS1 (1068.16 for (**a**) and 1071.03 (**b**)), respectively. For an explanation of the SN subdivision (Z1, Z2, Z3, and Z4), see the Figure 2 legend.

**Figure 5 plants-13-02785-f005:**
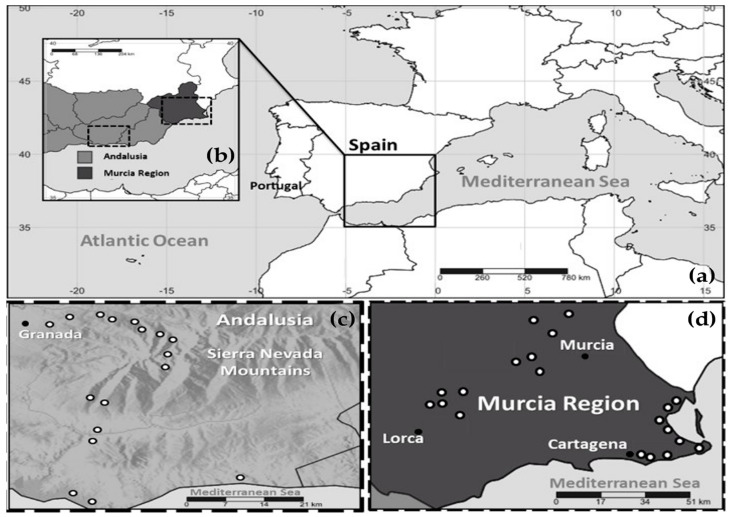
Geographical situation of the studied regions and sampled locations. (**a**) Map of the western Mediterranean area showing the southern part of Spain where the studied regions are located; (**b**) location of Sierra Nevada Mountains (SN) in Andalusia (left discontinuous square) and Murcia Region (MU) (right discontinuous square) in Spain; (**c**) distribution of the 17 sampled locations in SN, from which 84 samples were collected; (**d**) distribution of the 20 sampled locations in MU, from which 53 samples were collected.

**Figure 6 plants-13-02785-f006:**
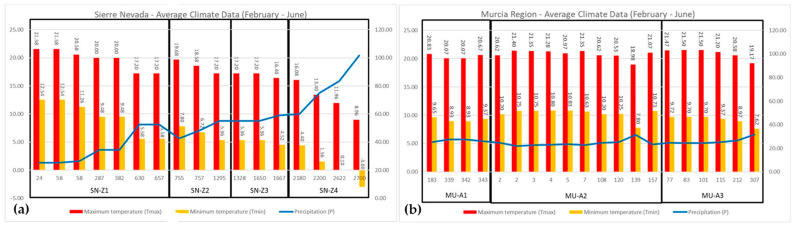
Climate diagrams of the two studied regions are depicted based on climate data from WorldClim–Global Climate Data [57], covering the period from February to June. The left axes indicate temperature (°C), while the right axes indicate precipitation (mm). Panel (**a**) illustrates the Sierra Nevada Mountains; Panel (**b**) represents the Murcia Region. The diagrams show the average maximum temperature (T max in °C), the average minimum temperature (T min in °C), and the precipitation (P in mm) during the assumed life cycle of the samples from each location. Locations are distinguished by their altitude in meters above sea level (m a.s.l.), and they are grouped into four altitudinal zones in the Sierra Nevada Mountains (SN-Z1, SN-Z2, SN-Z3, and SN-Z4) and three geographical areas in the Murcia Region (MU-A1, MU-A2, and MU-A3). In Panel (**a**), Tmax and Tmin decrease along the altitudinal gradient, while precipitation increases, whereas in Panel (**b**), Tmax, Tmin, and precipitation exhibit uniformity among sampling locations.

**Table 1 plants-13-02785-t001:** DNA polymorphism, genetic parameters, and statistics for each genotyped locus of *Funaria hygrometrica* populations across the Sierra Nevada Mountains (SN) and the Murcia Region (MU).

Locus	Region	Sequences Count ^a^	Amplification rate %	Group A Length (bp)/Count/Pairwise Identity %	Group B Length (bp)/Count/Pairwise Identity %	Total Pairwise Identity %	h	Hd	π%	N_m_	Tajima’s D
ITS1	SN	22/62	100	244–249/70/98.70	254–255/14/99.70	93.70	22	0.78	0.51	-	-
	MU	40	75.47	248/40/99.70	−	99.70	4	0.35	0.14	-	-
Total/ Alignment		124	90.51	281		95.40	25	0.67	0.40	3.02	−1.83 *
ITS2	SN	22/62	100	311–312/17/93.00	316/67/99.99	96.60	7	0.37	0.88	-	-
	MU	38	71.70	309/01/-	316/37/99.70	99.50	4	0.15	0.42	-	-
Total/ Alignment		122	89.05	331		97.60	10	0.31	0.75	4.97	−1.74 ^NS^
cpDNA	SN	22/62	94.05	778–779/12/99.80	794–798/67/99.50	99.10	25	0.72	0.20	-	-
	MU	21	39.62	795–797/20/99.90	802/01/-	99.80	5	0.35	0.02	-	-
Total/ Alignment		100	72.99	805		95.40	29	0.66	0.16	4.76	−2.18 **
mtDNA	SN	22/57	100	790–791/66/99.98	836/18/99.90	98.10	11	0.43	0.03	-	-
	MU	52	98.11	792/51/100	836/01/-	100.0	2	0.04	0.00	-	-
Total/ Alignment		136	99.27	837		98.6	13	0.65	0.02	2.71	−2.32 **

h: haplotypes number; Hd: haplotype diversity; π%: nucleotide diversity; Nm: gene flow; NS: not significant (*p* > 0.05); *: *p* < 0.05; **: *p* < 0.01. ^a^ For the SN samples, the highlands (SN-HL)/lowlands (SN-LL) sample numbers are shown.

**Table 2 plants-13-02785-t002:** Haplotype group combinations count for *F. hygrometrica* samples from the Sierra Nevada Mountains (SN) and Murcia Region (MU) populations. For SN, the values for each zone are also given. For an explanation of the SN subdivision (Z1, Z2, Z3, and Z4), see the Figure 2 legend.

Haplotype Combinations *	Count					
	Total for SN	SN-LL		SN-HL		Total for MU
		SN-Z1	SN-Z2	SN-Z3	SN-Z4	
AAAA	56	15	34	6	1	18
AA - A	5	3	2	0	0	20
A - - A	0	0	0	0	0	2
- - AA	0	0	0	0	0	2
- - A -	0	0	0	0	0	1
- - - A	0	0	0	0	0	9
BBBB	7	3	1	2	1	0
BAAA	2	0	1	1	0	0
ABAA	1	0	0	1	0	0
AABA	1	0	0	0	1	0
AAAB	1	0	0	0	1	0
- - - B	0	0	0	0	0	1
AABB	6	3	0	2	1	0
BABB	3	0	0	1	2	0
BBAB	1	0	0	0	1	0
BABA	1	0	0	1	0	0
**Total**	84	24	38	14	8	53

***** Each haplotype combination is represented by four letters; each corresponds to one locus (from left to right: ITS1, ITS2, cp, and mt). Missing loci are indicated by a hyphen.

**Table 3 plants-13-02785-t003:** Results of chi-square tests (upper line) and Fisher’s exact tests (bottom line) comparing *F. hygrometrica* populations from Sierra Nevada Mountains (highlands: SN-HL; lowlands: SN-LL) versus the Murcia Region (MU). A contingency table that reflected the geographical/altitudinal grouping of the samples versus haplotype Group A or B was used as an input. The haplotype groups were not randomly distributed in SN-HL vs. SN-LL or between SN-HL vs. MU.

Comparison	ITS1	ITS2	cpDNA	mtDNA
SN-HL vs. SN-LL	0.00038 * 0.00113 *	0.03397 0.04850	0.00057 * 0.00176 *	0.00014 * 0.00041 *
SN-HL vs. MU	1.2132 × 10^−05^ * 2.4519 × 10^−05^ *	0.01111 * 0.01984 *	0.00922 * 0.00378 *	2.9226 × 10^−07^ * 1.7011 × 10^−06^ *
SN-LL vs. MU	0.06551 0.17003	0.38055 0.64619	0.24415 0.56754	0.05116 0.06905

* Significant values (*p* > 0.05).

**Table 4 plants-13-02785-t004:** AMOVA *Φ* statistics of the four genotyped loci for all *F. hygrometrica* samples from Sierra Nevada Mountains (SN) and Murcia Region (MU) populations, contrasting different factors, geographical (SN vs. MU) and altitudinal (lowlands vs. highlands).

Contrast	Loci	*Φ* _CT_	*Φ* _SC_	*Φ* _ST_
Geographical	ITS1	0.02 ^NS^	0.18 *	0.20 *
	ITS2	0.07 ^NS^	0.09 ^NS^	0.15 *
	cpDNA	0.04 ^NS^	0.19 *	0.22 **
	mtDNA	0.07 ^NS^	0.32 *	0.22 **
Altitudinal	ITS1	0.41 **	0.00 ^NS^	0.40 **
	ITS2	0.22 **	0.03 ^NS^	0.24 **
	cpDNA	0.32 **	0.04 ^NS^	0.35 **
	mtDNA	0.47 **	0.16 *	0.55 **
Geographical and altitudinal	ITS1	0.03 ^NS^	0.04 ^NS^	0.07 *
	ITS2	0.14 *	0.01 ^NS^	0.15 *
	cpDNA	0.21 *	0.04 ^NS^	0.25 **
	mtDNA	0.27 *	0.16 *	0.39 **

*Φ*_CT_: between regions; *Φ*_SC_: within zones/areas of SN and MU, respectively; *Φ*_ST_: among locations; ^NS^: not significant (*p* > 0.05); *: 0.05 > *p* > 0.001; **: *p* ≤ 0.001.

**Table 5 plants-13-02785-t005:** Linkage disequilibrium (LD) among the four genotyped loci for *F. hygrometrica* populations across altitudinal zones in the Sierra Nevada Mountains (SN), southern Spain. Results are shown for all samples combined, as well as for samples below 1300 m a.s.l. (SN-Z1 and SN-Z2) and above 1300 m a.s.l. (SN-Z3 and SN-Z4). The *p*-values < 0.05 of the LD tests are presented in boldface.

Loci	All Samples				Samples below 1300 m a.s.l.				Samples above 1300 m a.s.l.			
	ITS1	ITS2	cpDNA	mtDNA	ITS1	ITS2	cpDNA	mtDNA	ITS1	ITS2	cpDNA	mtDNA
ITS1												
ITS2	0.00				0.00				0.04			
cpDNA	0.06	0.00			0.00	0.00			0.49	0.50		
mtDNA	0.00	0.00	0.00		0.00	0.00	0.00		0.31	0.21	0.22	

## Data Availability

All data are accessible through the following GenBank database–NCBI PopSet accessions: nrITS1, SN (442738645) and MU (1159852720); nrITS2, SN (442738667) and MU (1159852740); cp *rps*3-*rpl*16, SN (442738595) and MU (1159852771); and mt *rpl*5-*rpl*16, SN (433335516) and MU (1159852803).

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
