# Peer review of "Landscape Heterogeneity Drives Genetic Diversity in the Highly Dispersive Moss Funaria hygrometrica Hedw."

_plants, 2024, doi:10.3390/plants13192785_

Round 1
Reviewer 1 Report
Comments and Suggestions for Authors
Abstract: numeric data concerning genetic diversity, differentiation between populations, and selfing-rate are missing.
Keywords such as Murcia Region or Spanish Sierra Nevada Mountains could be substituted by some more valuable words related to systematics of the species, genetics or populations.
Introduction. It is better to mention research geography, research methods or the volume of research than the names of some researchers (rows 39-40 56, 59)
Some readers might be interested in systematic position of the species; in the present version of the article such information is missing.
Information about former molecular research of the species (or related species) is missing. Also selection of the markers is not commented.
AFLP is provided without explanation what does such abbreviation mean.
Results. Table 1. It is time consuming to analyse notes, somehow abbreviations SN-HL and SN-LL should be included inside the table, because later on in the description these abbreviation are used. In the table 2 the other type explanation is used as abbreviations SN-Z1 SN-Z2 SN-Z3 SN-Z4. Both information (LL, HL and Z types should exist in each table, etc.) – unification/clarification is required
Bigger letters of the figure 2 are required inside picture. There is no need to repeat in each row of the legends “haplotype”. The title should include the name of the species and countries: “Minimum spanning network, geographical distribution, and frequencies of the observed haplotypes are depicted” (row 201).
Table 3. Title: are mountains compared??? Logistic mistake
Table 4. Again – are regions compared???
The same concerns the titles of the fig. 3 and table 5.
Have the authors tried to separate longitudinal and latitudinal for Φ statistics (Table 4).
Indexes, like “Φ”, should be typed in italics
Figure 3 – bigger symbols for the titles of the axes should be used. The meanings of red, green, and white colours are not explained.
In the text SN and MU jargons should be updated by populations or other relevant word.
Table 5 should be in front of the fig. 3.
In the fig. 3 title is mixed with the descriptive text. Description should belong to the text rather than to the title. Paragraphs never end by tables or figures.
For “In resistanceGA analysis, altitude, geographical distance, land cover,
and human population density are crucial variables considered to assess how landscape
features influence genetic connectivity among populations.” references are required.
Rows 324-326 “Bioclimatic variables, including average annual temperature range (a) and precipitation of the wettest month (b), exhibit the highest association coefficients and lowest AICc (corrected Akaike information criterion) values for all loci (combined) and ITS1, respectively” should be outside the title of the table and some their numerical values should be provided.
Discussion. In this chapter, abbreviations SN and MU should be replaced by full names escaping confusion with indexes or DNA markers.
In the discussion topics related to the results should have references to the tables and figures.
English “...in this species the diversity is clearly structured according to altitude”... row 337
ΦCT – technical/typing errors
0.025 – 0.975 – should be without gaps (row 548)
The extent of genetic diversity of Funaria is not compared to the other molecular data of the same species also in the discussions no numerical comparisons of genetic diversity and differentiation are done in respect to the other moss species
It is not clear what research of Funaria has been done before present study, and if so, where this species has been investigated. The discussion is too general; it does not show relation to the other moss research.
References. Authors should check list of abbreviations of the journal titles – presently part of the titles are abbreviated and the other part – aren’t.
Comments on the Quality of English LanguageMinor revision is required. Mainly style - it might be valuable to diminish abbreviations.
Reviewer 2 Report
Comments and Suggestions for Authors
The author collective of the article deals with the genetic variability of the moss species in the area of ​​southern Spain. The authors use the analysis of selected loci of DNA from chloroplasts, mitochondria and the nucleus to assess genetic variability between and among regions. They use a lot of analytical tools to explain the connections with genetic variability. Therefore, there are many graphs and tables in the article, some of which are part of the appendices. The appendices could probably include also a graph of climate parameters (Fig. 6), but in the case of a journal that is published only electronically, it probably doesn't matter. It can be then little surprising that after so many analyzes the article does not end with a definite conclusion, but rather with the proposal of several hypotheses. This is not intended as a criticism of the conclusion of the discussion, rather a praise of the authors for being able to think critically about their results and admit different scenarios leading to the variability of the species in the studied area.
The main result of the study is genetically distinct samples from higher altitudes in the Sierra Nevada mountains. Therefore, I would suggest highlighting this already in the title. For example "Altitude drives genetic diversity..." instead of the overly general "Landscape heterogeneity drives". I would also limit the use of the influence of "landscape diversity" on genetic variability, or completely remove the "landscape heterogeneity" and "landscape diversity" from the article. When we realize that the samples were collected in the same types of environment (roadsides), i.e. the diversity of the surrounding landscape probably does not have too much influence on this, if we consider the easy spread of spores and the heterogeneity of the landscape (forests, mountains, other biotopes unsuitable for the studied species), it does not form barriers. In the article, you present the influence of the climate given by the different altitude, which is another independent variable compared to the heterogeneity of the landscape. I would like this to be clarified in the text. In the same way, there is no "landscape diversity" in graph no. 4, but temperature and precipitation.
Additional questions and minor recommendations:
I wonder if the genetically different plants from higher altitudes were also morphologically different. Did you microscope them? You mention the possible occurrence of cryptic species (for bryophytes, a problem currently being solved in science). However, morphological differences can also be caused by local conditions (morphological variability given by the environment, not genetically). But you can filter this out easily by having also cultured the samples in vitro. Thus, morphological differences with a justification for another taxon would appear if plants from higher altitudes showed certain differences even under the same growing conditions.
It is good that the collection locations are also listed. I assume that the samples from which the analyzes were performed are stored in the collection of one of the authors or an institutional herbarium. It is nice to include this information to the methodology (e.g. for possible later taxonomic revisions).
In the first sentence of the abstract, the information regarding "dispersion" is repeated twice. Abbreviate for example: Funaria hygrometrica, moss species with remarkable dispersal capacity, was selected as the ...
Round 2
Reviewer 1 Report
Comments and Suggestions for Authors
I think requested improvement has been done